# Sources of Light Density Microplastic Related to Two Agricultural Practices: The Use of Compost and Plastic Mulch

Benjamin van Schothorst [1], Nicolas Beriot [1,2,*], Esperanza Huerta Lwanga [1,3] and Violette Geissen [1]

[1] Soil Physics and Land Management Group, Wageningen University & Research, Droevendaalsesteeg 4, 6708 PB Wageningen, The Netherlands; benjaminvanschothorst@hotmail.com (B.v.S.); esperanza.huertalwanga@wur.nl (E.H.L.); violette.geissen@wur.nl (V.G.)

[2] Sustainable Use, Management and Reclamation of Soil and Water Research Group, Universidad Politécnica de Cartagena, Paseo Alfonso XIII, 48, 30203 Cartagena, Spain

[3] Agroecologia, El Colegio de la Frontera sur, 24500 Unidad Campeche, Mexico

* Correspondence: nicolas.beriot@wur.nl

**Abstract:** Microplastics (MPs) constitute a known, undesirable contaminant of the ecosystems. Land-based pollution is considered to be an important contributor, but microplastics in the terrestrial environment remains largely unquantified. Some agriculture practices, such as plastic mulch and compost application, are suspected to be major sources of microplastics as plastics are exposed to weathering or are present in organic fertilizers. The overall aim of this research is to bridge the terrestrial plastic contamination information gap, focusing on light density microplastics in two vegetable production systems in Southeast Spain and in the Netherlands. The selected farmer in Spain used plastic mulch for more than 12 years whereas the two farmers in the Netherlands annually applied 10 t ha$^{-1}$ compost for the past 7 and 20 years. Samples from two different depths were collected: 0–10 cm and 10–30 cm. High quality compost samples originating from municipal organic waste and from garden and greenhouse waste were obtained from two Dutch compost plants. All samples from both Spanish ($n = 29$) and Dutch ($n = 40$) soils were contaminated by microplastics, containing $2242 \pm 984$ MPs kg$^{-1}$ and $888 \pm 500$ MPs kg$^{-1}$, respectively. Compost samples from municipal organic waste ($n = 9$) were more contaminated than the ones from garden and green house wastes ($n = 19$), with, respectively, $2800 \pm 616$ MPs kg$^{-1}$ and $1253 \pm 561$ MPs kg$^{-1}$. These results highlight the need for studies focusing on the effects of microplastics in the environment and the need for monitoring campaigns and the implementation of thresholds to regulate the microplastic contamination.

**Keywords:** microplastics; plastic mulch; compost; vegetable production

## 1. Introduction

In 1997, Charles Moore, the man who sailed through the Great Pacific Garbage Patch, stated that "humanity's plastic footprint is probably more dangerous than its carbon footprint" [1]. In the 20-odd years that have followed, production of increasingly diverse plastics has continued to accelerate, standing in excess of 368 million tonnes per year since 2019 [2]. Moore's assessment about the danger of plastics is increasingly commonplace [3]. Institutions who understand the urgency to control the widespread plastic contamination implemented restricted measures on the plastic use. For instance, in 2019 the European Parliament adopted a ban on throwaway plastics to be implemented by 2021 [4].

After more than two decades, the adverse effects of plastic litter are still poorly understood. Plastic litter can be potentially harmful as (i) a physical threat to organisms, (ii) inherent toxicity, or (iii) a transport medium of other contaminants [5]. Plastic litter breaks down over time, and gradually fragments into smaller pieces, where it becomes more likely to infiltrate food webs [6,7]. Microplastic particles (MPs) are defined as plastics smaller than five millimetres [8], up until the micrometres range, being either directly

polluted as such, or produced from degrading plastics in the environment as secondary microplastics [9]. Research with regard to proliferation and impact of MPs has been primarily focused on marine, and, to a lesser extent, fresh-water environments, as opposed to the terrestrial environment [10]. About 4 to 23 times the amount of plastic waste released to oceans is estimated to be retained yearly in continental environments [11]. All the previous studies indicate the ubiquitous presence of microplastics in the terrestrial environment, with higher contents in urban areas and agricultural fields [12,13].

For most activities, plastic wastes should not enter the environment if handled and treated properly. However, in agriculture, plastics are put in direct contact with the environment and debris accumulates in the soil either because of the fragmentation of used plastics or the use of organic fertilizers contaminated with plastics [14].

Over time, plastics became the most economical solution to sustain high crop production for many agricultural activities. For example, plastic films can be used as mulch to increase the water use efficiency, to increase the soil temperature, and/or to control weed growth. Different types of plastic can be used for mulching. The most common is Low Density Polyethylene (LDPE) [15]. LDPE is a fully saturated polymer of hydrocarbons, which makes it highly resistant to weathering [16]. Consequently, LDPE mulch needs to be removed after harvest. This is a process during which LDPE debris accumulates in the environment. Some plastic producers have tried to improve the degradation processes of plastic to avoid plastic mulch removal and plastic debris accumulation. Pro-oxidant Additive Containing (PAC) plastics are polymers, mainly LDPE, which contain a pro-oxidant additive to enhance oxidation and photo-degradation [17]. In the presence of light and under aerobic conditions, PAC plastics degrade quickly into small pieces. Small, fragmented debris is more likely to be further degraded by microorganisms [18]. PAC plastics are also known as "oxo-degradable" or "oxo-biodegradable" [19]. However, when incorporated into the soil, the degradation process is minimized due to the absence of UV-light and PAC debris accumulates. Over the last few years, new mulching films that can be degraded by microorganisms in the soil have been developed [20,21]. They are usually sold as "biodegradable" mulch [22]. Biodegradable mulch can be made of a diversity of polymers [23], either bio-based, petroleum-based, or a blend of both.

Besides plastic films, sewage sludge and compost—which, among other organic fertilizers, are sources of nutrients in agriculture—can also be a source of microplastics. Sewage sludge contains microplastics from washed clothes, from personal care products, and from tires/road abrasion [24]. Composts become contaminated with microplastics when organic residues are collected along with plastic materials [25]. For example, agricultural plastics can be collected along with plant residues, while urban compost can be collected along with plastic bags, when garbage is not properly sorted. In the Netherlands, concerns have arisen about the possibility of plastic debris being present in industrial compost [26]. Fine particles can be unintentionally generated and sequestered through industrial scale composting processes, such as grinding [27]. Only a few studies have studied the sources of microplastics in the field and more assessments are needed to understand the input, accumulation, and transport of microplastics.

This research focuses on two farming systems representative of microplastic contamination in European agricultural fields. It examines the long-term annual use of compost in Dutch vegetable farms and of plastic mulch for vegetable production in Southeast Spain. The objectives of this study are (i) to quantify light density microplastics in agricultural soils under long-term plastic mulch and compost application in order (ii) to verify the compost applications as a source of microplastics pollution in agricultural soils where compost is applied.

## 2. Materials and Methods

### 2.1. Case Study

Two vegetable production systems were selected, one in Spain and one in the Netherlands (Table 1). In Spain, three different fields belonging to one farm were sampled in

the Murcia region with a crop rotation each, alternating lettuce, broccoli, celery, fennel, muskmelon, and watermelon. The soil was a Calcisol. A detailed record of plastic mulch application was available for the past 12 years. Initially, LDPE mulch had been applied for seven years (eight applications), after which PAC plastic was applied for four years (six applications) and biodegradable plastic was applied for one year (one application), for a total of 15 plastic mulch applications in the last 12 years.

**Table 1.** Sampling design description and abbreviations.

| Abbreviation | Location | Sample Type | Management |
|---|---|---|---|
| Sp | Murcia (Spain) | Soil (loam) | 15 plastic mulch applications in 12 years (8 LDPE+ 6 PAC + 1 biodegradable) |
| NL1 | Noordoostpolder (The Netherlands) | Soil (loamy sand) | 7 years of compost application |
| NL2 | Noordoostpolder (The Netherlands) | Soil (clay) | 20 years of compost application |
| Cm | The Netherlands | Compost | Organic materials from municipal waste |
| Cg | The Netherlands | Compost | Organic materials from green cuttings (garden and greenhouses) |

In the Netherlands, two distinct farms were sampled from the agricultural area of Noordoostpolder, which was reclaimed from the old Zuiderzee in the 1940s. The first farm's fields (NL1) mainly consisted of beach, loamy sands. The farmer cultivated tulip bulbs, onions, sugar beets, potatoes, chicory, and winter wheat in a 6-year rotational schedule. Compost application amounted to 10 t ha$^{-1}$ year$^{-1}$ for the last seven years. The first four years thereof compost from organic urban waste was applied, with the last three years shifting to composts based on green cuttings. The second farm's fields (NL2) consisted of heavy sea clay and provided a more typical type of soil for the area. The crop rotation schedule was potatoes every three years, wheat, and onions every six years, sugar beets every five years, and carrots every nine years. A total of 10 t ha$^{-1}$ year$^{-1}$ of high quality compost (certified so-called *Keurcompost* [28]) of mixed origins had been applied for the past 20 years.

*2.2. Sampling Design*

2.2.1. Sampling Soils in Spain and in The Netherlands

For both case studies, sampling was performed on a farm-to-farm basis, gathering soil from three fields per farm. Five randomised points were sampled at 0–10 cm and 10–30 cm (ploughing depth) for a total of 30 samples per farm. Soil was sampled with a manual auger, with a boring head volume of ~0.7 dm$^3$. The two depths were selected to assess the MP content in the root zone. Fields measured about 0.5 ha and were distinguished based on the cultivation and the application of plastic mulch or compost, respectively.

2.2.2. Sampling and Type of Composts in The Netherlands

Compost samples were provided by two composting companies in the Netherlands. The composts came from two different origins: source separated municipal organic waste (GFT *groente-, fruit-, en tuinafval* in Dutch) (Cm) and green waste from garden and greenhouse plant cuttings (Cg). Both types of compost were certified Keurcompost of class A, the highest quality class of compost. Class A compost does not contain more than 1% of materials >5 mm, no more than 0.05% of glass between 2–20 mm (bigger sizes have to be wholly absent), nor more than 0.05% of any other type of containment >2 mm [28]. MP loads are not subjected to scrutiny. As of 2016, 54% of green waste composts and 17% of municipal organic waste composts were labelled as class A [29]. This is a number which is increasing over time. In total, 20 subsamples were taken from each compost.

### 2.3. Microplastic Analysis

2.3.1. Microplastic Extraction with Flotation

The method of Zhang et al. (2018) for plastic extraction was adapted, separating buoyant and non-buoyant material [30]. Briefly, 10 g of dried 2 mm sieved soil or 5 g of dried compost were stirred in 30 mL of distilled water and centrifuged at 3000 rpm for 10 min. The supernatant was transferred onto a Whatman No. 42 filter paper (2.5 μm particle retention). Samples were refiled with distilled water, stirred again, and put in an ultrasonic bath to further break down soil aggregates. The samples were centrifuged again, and the supernatants were poured onto the same filters. The filters were then air dried for 24 h before microplastic identification and quantification were carried out.

2.3.2. Visual Microplastic Identification

All materials present on a filter were brushed carefully onto a glass plate and gathered in the centre of the plate while trying to avoid the superposition of particles. A stereo microscope (ZEISS Stemi 508) equipped with a digital camera (Leica) was used to take a picture of the particles with ×6 magnification. The glass plate was then put onto a hot plate at 130 °C for 10 s and a second picture was taken. The plastic particles were identified among other soil particles and organic matter by looking at their shape, colour, brightness, and response to heat [30]. Samples with too much organic matter, resulting in particle superposition, were not analysed. An example of a workable picture taken from a compost sample, before (Figure 1A) and after heating (Figure 1B), is offered in Figure 1. Plastic fragments were outlined using Adobe Photoshop CC 2018 (Figure 1C) before further analysing the pictures in ImageJ 1.52 (Figure 1D).

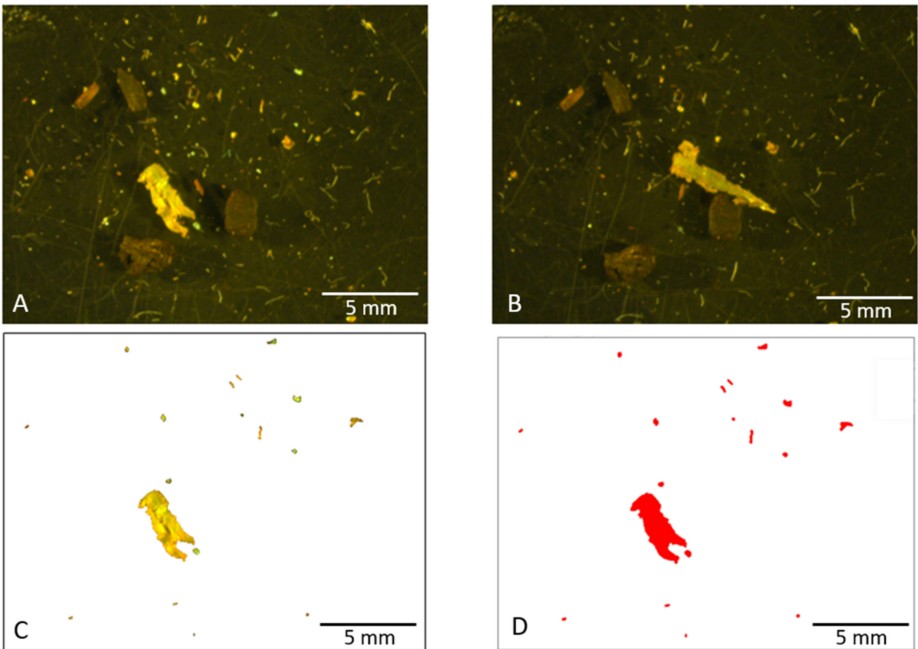

**Figure 1.** Example of the picture analysis procedure with a compost sample. (**A**) Initial picture of the material gathered on the glass slide after filtration. (**B**) The material after heating. (**C**) The visual identified particles. (**D**) Threshold mask for analysing particles in ImageJ.

2.3.3. Microplastic Particles Analysis with ImageJ

All pictures were analysed using the batch process of ImageJ 1.52. The pictures were first converted to 8-bit type and a threshold was applied before using the analysed particle functions. The number of particles per kg was estimated on the basis of total sample dry weight (10 g for the soil, 5 g for the compost samples). We detected particles bigger than

30 μm and smaller than 2 mm. MP were classified into size fractions, <100 μm, 100–200 μm, 200–300 μm, 300–500 μm, > 500μm, by calculating the square root of the area [8].

### 2.3.4. Plastic Confirmation with Fourier Transform Infrared Spectrometer

Millimetre ranged particles were picked up for analysis with a Varian 1000 FTIR (Fourier transform infrared) spectrometer in order to identify the type of plastics. Three MPs found in Dutch agricultural soil and five MPs found in compost were analysed. The spectrometer produced absorbance spectra ranging from 4000 $cm^{-1}$ to 750 $cm^{-1}$ with a resolution of 4 $cm^{-1}$. The spectra were compared with the polymer library (HR Hummel Polymer and Additives). Reference spectra for Polyethylene (PE) and Polypropylene (PP) are shown in Figure S1.

### 2.3.5. Microplastics Input Calculations Per Compost and Per Plastic Mulch

Yearly estimations of the plastic input per ha were calculated using the data provided by the farmers (Table 1). These estimations were then converted into MPs per kg of soil per year, considering the 30 first centimetres of soil and a soil density of 1400 kg $m^{-3}$ and compared to the measured MPs content in the soil. For a Spanish farm, we divided the area of plastic mulch applied per ha in the past 12 years with the soil density to estimate the area of plastic mulch applied per kg of soil in the past 12 years. For the Dutch farms, we multiplied the average MPs content (number and area per kg) measured in all the compost samples by the average compost application and the number of years of applications to estimate the number and area of plastic input per kg of soil.

### 2.4. Statistical Analysis

A normality test was carried out for every sampling set, determining cumulative distribution function (CDF), expected values, and z-values, which were subsequently plotted in a Q-Q plot to test for normality. The data was normally distributed. We compared the means of each sample with a Kruskal–Wallis and Wilcoxon test implemented in R version 3.6.1. Particle size distributions were calculated with the geom_density function and were compared with a Kolmogorov-Smirnov test. Graphics were made with the function ggplot in R.

## 3. Results

Microplastics were found in all samples. A summary of the contents can be found in Table 2. The Fourier transform infrared (FTIR) spectroscopy analysis confirmed the presence of PE and PP plastics in compost and confirmed the presence of PE in agricultural soil (Figure S1).

**Table 2.** Number of analysed samples and average microplastic contents for all origins (Spanish Scheme 1. NL2), Dutch compost (Cm, Cg)), and both soil depths (0–10 cm and 10–30 cm).

| Sample Origin | Depth | Num. Samples | Num. MPs $kg^{-1}$ | Area MPs [$mm^2$ $kg^{-1}$] |
|---|---|---|---|---|
| Sp | 0–10 | 15 | 2302 ± 937 | 215 ± 89 |
| Sp | 10–30 | 14 | 2179 ± 1063 | 184 ± 120 |
| NL1 | 0–10 | 11 | 903 ± 430 | 95 ± 60 |
| NL1 | 10–30 | 11 | 848 ± 586 | 104 ± 156 |
| NL2 | 0–10 | 8 | 650 ± 245 | 67 ± 86 |
| NL2 | 10–30 | 10 | 1107 ± 587 | 99 ± 78 |
| Cm | - | 9 | 2800 ± 616 | 212 ± 51 |
| Cg | - | 19 | 1253 ± 561 | 137 ± 157 |

### 3.1. Number of MPs in Compost, Dutch, and Spanish Soils

Spanish agricultural soils (Sp) contained significantly more MPs than the Dutch soils with respective averages of 2243 ± 983 MPs $kg^{-1}$ and 888 ± 500 MPs $kg^{-1}$. We did not

observe significant differences between the plastic content in the top soil (0–10 cm) and the deeper soil (10–30 cm) except in the farm NL2, where more MPs were observed in the deeper layer (*p*-value = 0.036). As pertaining to composts, the compost from municipal organic waste (Cm) contained significantly more MPs than the compost from green cuttings (Cg) with 2800 ± 616 MPs kg$^{-1}$ and 1253 ± 561 MPs kg$^{-1}$, respectively (Figure 2).

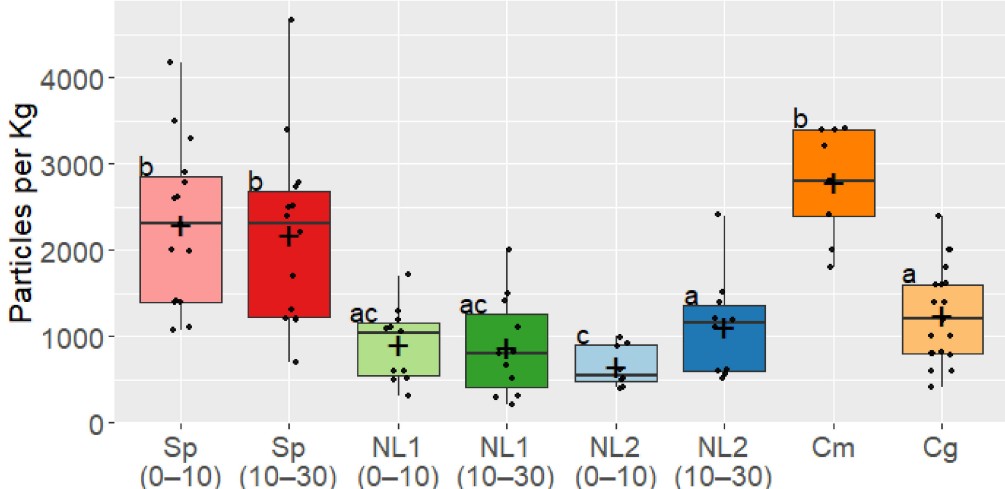

**Figure 2.** Microplastic content in agricultural soil of the Spanish farm (Sp) and Dutch farms (NL), for both the top soil (0–10 cm) and the deeper depth (10–30 cm) and for compost samples from municipal waste (Cm) and from green cuttings (Cg). The box plot (horizontal lines) represents content for at least 25%, 50%, and 75% of the samples. The vertical black line ends represent the minimum and maximum values. The cross represents the average content of any given sample group. The dots represent individual measurements. Treatments that do not share letters are significantly different from each other (Wilcoxon comparison at *p* < 0.05).

*3.2. Size Comparison of MPs between Locations*

The MPs found in NL1 were significantly bigger than the MPs in NL2 (Figure 3). Additionally, in NL1, bigger MPs were found in the top soil depth (0–10 cm) than in the lower soil depth (10–30 cm). The MPs' size distribution was similar for both compost origins.

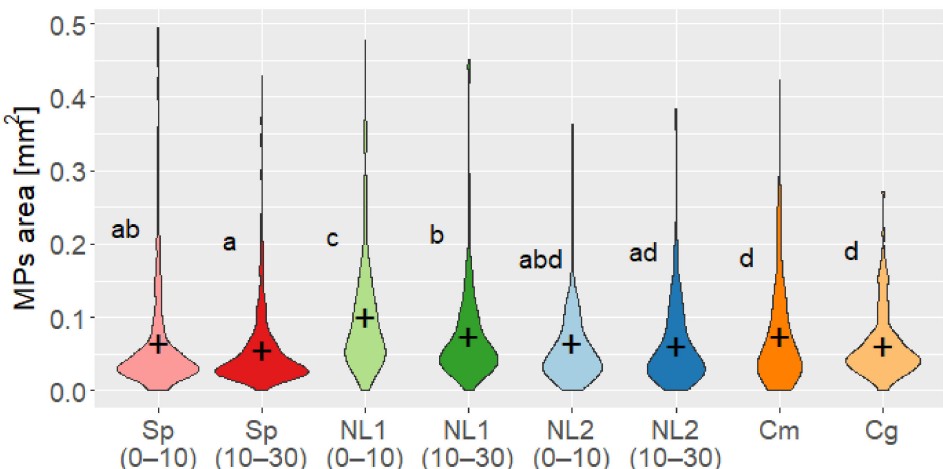

**Figure 3.** Microplastic particles' area distribution in agricultural soil of the Spanish farm (Sp) and Dutch farms (NL) for both the top soil (0–10 cm) and the deeper depth (10–30 cm), and for compost samples from municipal waste (Cm) and from green cuttings (Cg). Only particles < 0.5 mm$^2$ are plotted for a better visualization. The cross indicates the mean particle area. Treatments that do not share letters are significantly different from each other (Kolmogorov-Smirnov test at *p* < 0.05).

### 3.3. Estimating the Source of MPs in the Spanish and Dutch Soils

The farmer in Spain recorded 15 plastic mulch applications in the past 12 years. At each plastic mulch application, about half of the field was covered. This represents ~7.5 ha of plastic mulch per ha of soil in the past 12 years (Table 3). We do not know the efficacy of plastic removal after harvest. We can estimate without a removal process for the top 30 cm of soil, a total plastic input of ~$1.8 \times 10^4$ mm$^2$ kg$^{-1}$ in the past 12 years, which represents an annual input of ~$1.5 \times 10^3$ mm$^2$ kg$^{-1}$ year$^{-1}$. The microplastic content in the Spanish farm (200 mm$^2$ kg$^{-1}$) that we measured represents ~1% of the plastic that was applied.

**Table 3.** Estimation of the plastic input from a plastic mulch application in the Spanish farm soil. The calculation does not take into account the plastic removal (unknown rate) after the harvest.

| Plastic Mulch Application | Plastic Mulch Applied Per Hectare [mm$^2$ ha$^{-1}$] | Plastic Mulch Applied Per Soil Mass in the Top 30 cm [mm$^2$ kg$^{-1}$] |
|---|---|---|
| Total (12 years) | $7.5 \times 10^6$ | $1.8 \times 10^4$ |
| Average per year | $6.25 \times 10^5$ | $1.5 \times 10^3$ |

The known application rate of composts equals 10 t ha$^{-1}$ year$^{-1}$ in Dutch soils. With an average content of ~2026 MPs kg$^{-1}$ for both compost types, a soil density of 1400 kg m$^{-3}$ and a ploughing depth of 30 cm, we calculate an annual input of ~5 MPs kg$^{-1}$ year$^{-1}$ in the top 30 cm of soil (Table 4). A similar calculation for an average plastic area of 175 mm$^2$ kg$^{-1}$ gave an annual input of 0.42 mm$^2$ kg$^{-1}$ year$^{-1}$. Compared to the seven years (NL1) and 20 years (NL2) of application, the average plastic content we measured in compost can explain <10% of the plastic content in the Dutch soils.

**Table 4.** Estimation of the plastic input from the compost application in the Dutch farms (NL1 and NL2).

| Plastic Mulch Application | Compost Application [t ha$^{-1}$] | Number of MPs Imported * [MPs kg$^{-1}$] | Area of MPs Imported * [mm$^2$ kg$^{-1}$] | Number of MPs Measured [MPs kg$^{-1}$] | Area of MPs Measured [mm$^2$ kg$^{-1}$] |
|---|---|---|---|---|---|
| Total NL1 (7 years) | 70 | 35 | 3 | 886 | 100 |
| Total NL2 (20 years) | 200 | 100 | 8.3 | 904 | 83 |
| Average per year | 10 | 5.0 | 0.42 | 66.3 | 6.78 |

\* estimation with an average MPS content in compost of 2026 MPs Kg$^{-1}$ and 175 mm$^2$ kg$^{-1.}$

## 4. Discussion

### 4.1. Microplastics in Compost

It was long assumed that organic fertilizers would be a vector of MPs [31]. However, until now, very few studies analysed the plastic content in compost. Weithman et al. (2018) [8] identified 24 MPs kg$^{-1}$ of size 1 mm to 5 mm in German compost from municipal organic waste and green clippings. A more recent study measured ~2400 $\pm$ 358 MPs kg$^{-1}$ in composts in the Zhejiang Province (China) for MPs from 50 μm to 5 mm [27]. These two results are very similar to the ~21 $\pm$ 31 MPs kg$^{-1}$ for MPs between 1 mm and 2 mm and 1750 $\pm$ 930 MPs kg$^{-1}$ for MPs between 30 μm and 2 mm that we found in the Dutch compost.

We found about twice as many MPs in Cm than in Cg, suggesting that compost from municipal organic waste is more subject to plastic contamination than compost from garden and greenhouses green cuttings. This was expected because the volume of organic material compared to plastic packaging is higher for green cuttings than for municipal organic waste. Zee and Molenveld (2020) [25] studied a GFT compost in the Netherlands similar to Cm. They identified flexible plastic packaging (film) to be the main source of plastic in municipal compost. The difference between Cm and Cg shows the need for more MPs assessments in compost to (1) warn farmers or other compost users about the MPs contamination and (2) establish priority contamination management strategies.

Zee and Molenveld (2020) [25] and Gui et al. (2021) [27] both showed that Polyester, Polypropylene (PP), and PE accounted for the majority of the total amount of MPs in the compost [25,27]. This is consistent with the distribution of polymers in the plastic demand in Europe for packaging and households [32]. Reducing the plastic content in compost will require efforts to reduce plastic contamination of the organic waste as well as innovative composting processes to decrease the MP content in the compost.

### 4.2. Accumulation of Microplastics in Soils

Considering a similar yearly input of plastic through the compost and 13 more years of compost application for NL2 than for NL1, we would expect to find more MPs in NL2 than in NL1. However, we observed similar numbers of MPs in both farms, with smaller MPs in the soil of NL2 than in NL1. We could hypothesise that the longer history of plastic application for NL2 also represents more time for the degradation processes to reduce the MPs to a size smaller than 30 μm, making the plastic below our detection limit. Additionally, transport processes, such as wind erosion or water runoff and infiltration, could transport MPs away from the top 30 cm of soil [11,14]. We also observed more plastic in the deeper soil than in the top soil in NL2 and a similar amount of plastic at both depths for NL1, with smaller MPs in the deeper soil. These two results suggest the transport of small MPs deeper in the soil, with bigger particles staying in the top soil in case of NL1 but not in NL2 because the particles are, on average, smaller and because of the longer history of MPs contamination. Hydrological processes are projected to preferably transport intermediate particle sizes (10–100 μm), whereas bigger particles are more rapidly trapped [33,34]. Additionally, MPs can also be ingested and transported by terrestrial fauna [35–37]. For example, earthworms can ingest microplastics <250 μm, leading to their transport to deeper soil depth [38]. However, the fields that were sampled were ploughed every year, reducing the difference of MPs content between the layers. There may be other historical factors in play that we did not account for, possibly even a difference in the source material and its plastic contamination levels. We highlight the need for more studies to understand the MPs' transport processes and predict the MPs' accumulation.

It is worth noticing that the plastic content in the compost applied in past years explains less than 10% of the plastic content found in the Dutch soils. Other plastic sources can be considered. First, it is possible that the plastic content in the previous years was higher or that sewage sludge was applied in the previous years. Sewage sludge is a known vector of MPs [24]. For example, sewage sludge application was estimated to be responsible for the input of ~$10^8$ MPs ha$^{-1}$ year$^{-1}$ in cereal fields [39]. Besides organic fertilizers, another important source of microplastic could be wild dumping [14] and atmospheric deposition [40]. Studies account for microplastic atmospheric deposition rates, ranging from 0 to 11 130 particles·m$^{-2}$ d$^{-1}$ [41]. The average deposition of 76 MPs m$^{-2}$ d$^{-1}$ measured in an open field in the Hamburg metropolitan area [42], is more important than the estimated ~6 MPs m$^{-2}$ d$^{-1}$ input from the compost we measured. At a different scale, grazing animals can ingest plastic debris in agricultural fields and defecate microplastics in another area [43]. For example, a herd of sheep could be responsible for the transport of ~1 MPs m$^{-2}$ d$^{-1}$ [44].

The MP content in the Spanish soil was comparatively higher than in the Dutch soil and was similar to the one found in a previous study in the same area for similar agricultural managements [44]. The conventional plastic mulch is removed from the fields after the harvest. Therefore, the quantity of plastic measured in the soil accounted for only 1% of the plastic mulch area applied in the field in the last 12 years. In addition to the plastic removal, we can expect that wind erosion and water runoff will transport plastic debris away from the field and be partly responsible for the low recovery. Degradation, especially in case of biodegradable plastics, is also to be considered to explain the low recovery. On the other hand, since compost application does not fully explain the MP content in the Dutch farm, we can expect that other sources, as mentioned previously, also contribute to the accumulation of plastics in the soil.

This research adds to the growing consensus about MP abundance in agricultural soil and organic wastes [13], with 100% of samples contaminated. More assessments are needed to reveal the contribution of the different pathways of microplastics in the soil and to make a clear link between specific agricultural management and plastic contamination.

*4.3. Limitations of the Plastic Extraction and Identification Method*

The floatation extraction and visual identification of MPs successfully allowed us to estimate MPs content, area in soil, and compost samples, and has been validated in numerous studies [45]. This method, developed by Zhang et al. (2018), is a cheap and relatively fast method compared to spectral methods (e.g., Raman or Fourier transform infrared) [46,47]. However, we identify some limitations.

There is an urgent need for consistency and standardized protocols for identifying and quantifying microplastics. The wide variety of methods makes comparisons of large-scale results difficult [48,49]. For instance, particle counts are often listed in items $m^{-2}$ [50], or even MPs $km^{-2}$ in aquatic environments [51]. In accordance with the suggestion of Horton et al. (2017) [11], the units which are presented in this study are a unit per mass of soil, namely MPs $kg^{-1}$. The size considered in the analysis also has to be clearly defined. For example, Harms et al. (2021) [52] reported $3.7 \pm 11.9$ MPs $kg^{-1}$ in arable lands in Germany for particles between 1 mm and 5 mm. This is much lower than the thousands of particles we measured, but comparable to the amounts we find for particles between 1 mm and 2 mm in the Spanish and Dutch soil, respectively ~$21 \pm 18$ MPs $kg^{-1}$ and ~$8 \pm 13$ MPs $kg^{-1}$. On the other scale side, most methods studying microplastics have the lower limit of detection around 20–30 μm [53]. The detection of smaller plastics' particles remains a challenge [54].

Additionally, pre-processing to reduce the organic matter content has to be further implemented [55]. Soil organic matter is tricky to separate from microplastics, and is known to potentially hide plastic fragments in visual analyses, as well as to distort plastic signals in FTIR typology measurements [26]. Inadequate involvement of separating procedures has rendered many compost samples useless in this study. It is not advisable to apply the method of Zhang et al. (2018) for composts, without introducing additional steps like sieving to isolate a size fraction [8,50], or digestion to remove biogenic materials [56].

Furthermore, the method we used only applies for light density plastic that can be visually differentiated from soil particles after the heating treatment. The recognition remains arbitrary and invites human bias. FTIR analysis clarified the presence of PE in Dutch soils and both PE and PP in Dutch composts, but no exhaustive typology determination was deemed possible. Other plastics, which are not floating in water, are also used in agriculture and were not detected in our approach. Therefore, the results likely underestimate the total microplastic content.

**5. Conclusions**

This study confirms the ubiquity of microplastic contamination in agricultural soils. In two distinct agricultural systems, plastic mulch and compost appear to be major sources of plastic contamination. More emphasis on identifying contributors to agricultural soil pollution is warranted, in order to quantify and ultimately regulate the total influx of plastic pollution to agricultural soils. Results of this research indicate such quantification and regulation is urgently needed. Plastic mulch may be used and compost may be applied to continue to utilize the services they provide, but monitoring campaigns and quality standards have to be implemented to control the microplastic accumulation in soil. In the same way, the plastic content in composts should be monitored and kept down to a low level, ideally with maximum threshold values. Additional studies about the effects and transport of microplastics in the ecosystem are needed to set such threshold values.

**Supplementary Materials:** The following are available online at https://www.mdpi.com/article/10.3390/environments8040036/s1. Figure S1: FTIR analysis of 8 tested particles from composts and Dutch soils.

**Author Contributions:** Conceptualization, E.H.L., N.B., V.G., and B.v.S. Investigation, B.v.S. and N.B. Formal analysis, B.v.S. and N.B. Writing—original draft preparation, B.v.S. Writing—review and editing, B.v.S., N.B., E.H.L., and V.G. Supervision, E.H.L. Project administration, V.G. Funding acquisition, V.G. All authors have read and agreed to the published version of the manuscript.

**Funding:** This research was funded by the European Commission Horizon 2020 project Diverfarming, grant number 728003.

**Data Availability Statement:** The raw data and the R script for calculations and plots are available on the GitHub page https://github.com/NGBeriot/MPs_soil_compost (last accessed on 16 April 2021).

**Acknowledgments:** We are thankful for the contribution of the farmers involved in the study. We would also like to thank the compost companies for their collaboration. We acknowledge Carolien Kroeze from the Water Systems and Global Change of Wageningen University for her help and good advice.

**Conflicts of Interest:** The authors declare no conflict of interest.

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
