# Peer review of "Sources of Light Density Microplastic Related to Two Agricultural Practices: The Use of Compost and Plastic Mulch"

_environments, doi:10.3390/environments8040036_

Round 1
Reviewer 1 Report
Comments on “Sources of light density microplastic related to two agricultural practices: the use of compost and plastic mulch” by van Schothorst et al. for publication in Environments.
The manuscript reports on results concerning the extraction, determination and quantification of microplastic particles in soils submitted to two agricultural practices, compost amendment and mulching. The methodologies employed are not new and the discussion sounds sometimes like a literature review. However, the results provided reinforce the databases regarding the amounts of MP introduced in soils through widespread agricultural practices. This work will then probably be of interest to the scientific community. Moreover, it is well written and synthesized. For these main reasons and following the comments detailed below, I think that this paper reaches the publication standards of Environments and needs only minor improvements.
Abstract
Line 19. “High”: the capital letter is missing
Line 22. Suggest to replace “polluted” by “contaminated”
Introduction
Clear and well-structured introduction, supported by a diverse and up-to-date bibliography.
Materials and methods
2.1 Case study. Precise the surfaces of the sampled sites.
2.2. Sampling design. I suggest to add some rationale about the depths chosen. Moreover, what was the mass (or volume) of sampled soils and composts? Were they composite samples?
2.3. Microplastic analysis.
Line 146: the porosity of the filter paper must be specified.
Lines 151-152: did the authors verify that all the MP particles present on the filter were effectively recovered and counted?
Figure 1. A scale should be added on the pictures.
Lines 170-171: why did the authors choose to express their data using a number of MP particles per soil mass rather than a mass of MP particles per soil mass? If the data concerning the mass of the MP particles retrieved in soils are available, I suggest to add such values.
2.4.1. Calculation and statistical analysis. There is no section “2.4.” Please, verify the section numbering.
Results
3.1. MPs in compost, Dutch and Spanish soils.
Line 192. Remove the question mark.
Table 2: for a better readability, the authors should provide details concerning the statistical differences in MP contents between soils (and depths) and between composts (for instance using letters to indicate the significant differences)
Figure 2. The meaning of the cross is unclear in the caption. The Kruskal-Wallis test used here should be cited in the M&M section devoted to the statistics.
Figure S1. The size distribution is based on particle area (mm²). Why did not the authors show the data in µm (as proposed lines 172-174)? In addition to the amounts of plastics, their size distribution is a relevant information, especially with regard to the mobility and toxicity of MP (see also my comment on the discussion). These data could therefore be presented in the body of the manuscript.
3.2. Estimating the source of MPs in the Spanish and Dutch soils
Line 221. precise in Dutch soils.
Line 223. remove the capital letter from “Kg”
Discussion
The discussion is well written and supported by a dense literature. However, it seems to me that some results of the paper should be more explicitly discussed, e.g., the size distribution of the MP particles found in both soils and compost. Likewise, two different composts were studied but the effect of their origin/composition on their MP contents is not discussed either. Please, consider to add such elements in the discussion.
4.3. Remobilization of soil-bound microplastics.
Lines 315 and 320. Check citations.
Data Availability Statement. The link provided to get access to the raw data and the R scripts does not work.

Reviewer 2 Report
I thank the authors for this interesting paper. Overall I thought the promise of the paper was really interesting but I thought there was too much time spent on the discussion and not enough of describing the data and discussing the data presented within the article. Below are some key points and typos.
I will add that I very much enjoyed reading the introduction.
Line 51: "4-23" no sure what this means
Line 79: "diversity of polymers". This could be expanded because a bio-based polymer can still be PE. Chemically the end product is the same though they may have come from differing sources.
Line 82: "spelling of nutrients"?
Line 91: "Only a few"...
Line 105: what is going on with the full stops?
Line 107: tense - "had" been applied
Line 108: what type of biodegradable plastics? PAC/cellulose based?
Line 121 Table 1: abbreviation = Cm
Could the table be expanded to included the yearly estimations of plastic input per ha or the calculation of MPs per kg soil per year (as described in lines 185-188). This would make it much easier for the reader to cross compare with the values that you observed and describe so well in Table 2. Alternatively it could be a separate table and support section 3.2 (lines 215-277).
Line 175: random bracket
Line 192: random question mark, the p-values are missing as is the concentration of MPs for Spanish soil.
Line 196: was this observation significant? what was the p-value?
Line 198: the word "respectively" should go before Figure 2 reference on Line 199.
Line 200:What size were the particles?
Line 200-202: this could be expanded. I think there is interesting data here and it is squeezed away apologetically. What a shame. If I understand from your S2, you only found one sample of PP and everything else was PE? How does this link in with what was applied and the source?
204: Table legend doesn't match with table in that you refere to Spanish soil, Dutch soil etc but inside the table sample origin is Sp, NL1 etc.
Figure 2: Your statistics and what is significantly different is difficult to understand here - there is no description of what a, b etc refer to. Presumeably a=no significant difference, but this is not stated anywhere.
Line 220: What has happened to the plastic that is no longer measured? Has it been removed, degraded?
Section 221-227: It is not clear the point that you are trying to make. Neither is the data presented in the same was as it was for the Spanish soil. There seems so much data that could be discussed here and it is hurried through. A table showing total plastic applied or plastic applied per year together with plastic content at time of sampling....?
Line 232: spelling "from"
Line 238: you showed majority PE. Could you link the Gui observations with your data?
Section 4.2 (Lines 243 to 259). It would be helpful to the reader if you were consistent in presentation of Spanish data followed by Dutch data.
Line 248: mention of sewage sludge. Do you know if this was applied and if so when?
Line 256: If I understand your statement here, only 1% of plastic applied is detected in soil. What has happened to 99% of applied plastic? If I have got this wrong then a table of numbers would very much help.
Line 281: A lengthy discussion about standardized protocols leads to the presentation of data from the soil in this study that was not presented in the results (line 291). Where has this data come from?
Line 315: something not quite right with referencing
Supplementary figure 1:
How has %density been calculated? could you split Sp(0-10) and Sp(10-30) etc as som of the data is a little lost.
Supplementary figure 2:
units on axes are missing
labelling is very confusing. Perhaps a layout of spectra with standards, followed by Sp (0-10), Sp(10-30) etc as used throughout the rest of the paper would be helpful?
I believe the authors have some very interesting data but it was difficult to visualise it.
Round 2
Reviewer 1 Report
The authors have answered the questions and addressed some of the weaknesses pointed out by the reviewers.
The paper reaches the publication standards of Environments and will probably be of interest to the scientific community.
Author Response
We are very thankful to the reviewer, whose comments and suggestions contributed to reach the publication standards of Environments.
Reviewer 2 Report
I thank the authors for their responses and congratulate them on the huge amount of work undertaken. I particularly enjoyed the additional tables as they enhanced the story a great deal. I have very much enjoyed reading the manuscript.
Two comments below:
Figure 2 legend. You stated in your response to reviewer, but it seems it got missed in the version submitted:
The legend description was :
“Different letters indicate significant differences after a Wilcoxon comparison at p < 0.05”
We changed it to :
“Treatments that don’t share letters are significant different from each other (Wilcoxon comparison at p < 0.05)”
Spelling of municipal on line 274.
Author Response
Thank you to the reviewer for his nice comments.
We changed the legend in Figure 2 and Figure 3 to add "Treatments that don’t share letters are significant different from each other"
We corrected the spelling of municipal line 276.